# ABVENTURE-P pilot trial of physical therapy versus standard of care following ventral hernia repair: Protocol for a randomized controlled trial

**Stephanie Di Stasi**[1,2]*, **Ajit M. W. Chaudhari**[2,3], **Savannah Renshaw**[4], **Lai Wei**[5], **Laura Ward**[6], **Elanna K. Arhos**[2,3], **Benjamin K. Poulose**[4]

1 Division of Physical Therapy, School of Health and Rehabilitation Sciences, Ohio State University, Columbus, Ohio, United States of America, 2 Sports Medicine Research Institute, Ohio State University Wexner Medical Center, Columbus, Ohio, United States of America, 3 School of Health and Rehabilitation Sciences, Ohio State University, Columbus, Ohio, United States of America, 4 Center for Abdominal Core Health, Division of General and Gastrointestinal Surgery, Department of Surgery, Ohio State University Wexner Medical Center, Columbus, Ohio, United States of America, 5 Department of Biomedical Informatics, Center for Biostatistics, Ohio State University College of Medicine, Columbus, Ohio, United States of America, 6 Outpatient Rehabilitation Services, Ohio State University Wexner Medical Center, Columbus, Ohio, United States of America

* stephanie.distasi@osumc.edu

**Data Availability Statement:** No datasets were generated or analysed during the current study. All

## Abstract

Hernia disease is one of the most common reasons patients seek surgical treatment, yet nearly 1 in 4 patients seeking ventral hernia repair in the United States suffer from chronic pain, disability, and diminished physical activity. The relationships between the anterior abdominal wall, lower back, diaphragm, and pelvic floor are critical in providing function and quality of life, yet management of hernia disease has been limited to surgical restoration of anatomy without taking into consideration the functional relationships of the abdominal core. Therefore, the primary goal of this study is to evaluate the feasibility of implementing physical therapy targeted to improving stability and function in this population. A secondary goal is to estimate whether pre-operative abdominal core function predicts responsiveness to physical therapy. This study is a registry-based randomized controlled trial (NCT05142618: Pilot Trial of Abdominal Core Rehabilitation To Improve Outcomes After Ventral Hernia Repair (ABVENTURE-P)). All participants will be randomized to one of two post-operative treatment arms: standard of care plus up to 16 sessions of physical therapy, or standard of care alone. Primary timepoints include pre-operative (baseline) and ten weeks after surgery, with intermediate or secondary timepoints 30 days, 6 months, and 1 year post-operative. At each timepoint, participants will undergo functional and patient-reported outcome testing. We will also collect data on retention rate and treatment adherence. An intention to treat approach is planned for all analyses, using all participants who were randomized and have available data at the 10-week timepoint. This is a pilot and feasibility trial, hence our goals are to establish safety and initial efficacy of the PT intervention, retention and adherence to both PT and control arms, whether pre-operative abdominal core function predicts

relevant data from this study will be made available upon study completion.

**Funding:** This study is funded by the National Institute of Diabetes, Digestive and Kidney Diseases (https://www.niddk.nih.gov) grant R01DK131207 (SD, AMWC, BKP MPI's). This project also receives in-kind administrative support by the Abdominal Core Health Quality Collaborative (ACHQC) and by The Ohio State University Center for Clinical and Translational Science, which receives financial support from the National Center for Advancing Translational Sciences (NCATS) (UL1TR002733). Neither NIDDK nor NCATS assisted in the design of the trial and will not contribute to data collection and management, analysis and interpretation, dissemination of trial findings, and does not have authority over these activities. The content is solely the responsibility of the authors and does not necessarily represent the official views of the National Institutes of Health. The funders had and will not have a role in study design, data collection and analysis, decision to publish, or preparation of the manuscript.

**Competing interests:** I have read the journal's policy and the authors of this manuscript have the following competing interests: Benjamin K. Poulose receives salary support as Vice President of the Board of the Abdominal Core Health Quality Collaborative and is an equity holder in EndoEvolve, LLC. This does not alter our adherence to PLOS ONE policies on sharing data and materials.

responsiveness to PT, and to collect a large enough sample to power a future definitive multi-center randomized controlled trial.

## Introduction

Hernia disease is one of the most common ailments that disrupts the function of the abdominal core, with annual surgical repairs nearing 350,000 in the United States alone [1]. Recurrence rates for ventral hernia repair are as high as 32% [2], increasing both health care related costs to patients and risk of a transition to chronic post-operative pain. Given the high percentage of hernia recurrences after surgery, as well as the current opioid epidemic faced by the United States, there is a significant need for drug free treatments targeting abdominal core health to achieve optimal outcomes after ventral hernia repair.

The abdominal core, which consists of the anterior abdominal wall, spine, flanks, diaphragm, and pelvic floor, supports multiple critical functions important to activities of daily living, from respiration to personal hygiene to independent mobility. We define *abdominal core health* as encompassing the function, stability, and quality of life involving the abdominal core. The current lack of understanding of how hernia disease negatively impacts both abdominal core health and patient well-being precludes the development of more effective treatment.

Traditionally, management of hernia disease has been limited to the surgical restoration of the abdominal wall anatomy without taking into consideration the functional relationships of the abdominal core. Physical therapy (PT) can influence all inputs, control patterns and functions of the abdominal core that leads to improved quality of life. Physical therapy targets specific deficits in muscle performance and disrupted movement patterns, effectively reducing pain and improving quality of life and activity participation in a variety of orthopedic patient populations [3–5].

Over recent years, the concept of the abdominal core as a dynamic, functional unit has increasingly gained acceptance among medical professionals treating hernia disease [6]. However, this recognition has not yet led to inclusion of physical therapy as the standard of care following ventral hernia repair (VHR) and none of the 23 randomized controlled trials performed in this clinical area have considered the impact of intervention on core stability or have evaluated PT as a necessary adjunct to recovery [7]. Therefore, there is a critical need to establish (a) whether post-operative rehabilitation can augment outcomes of VHR, and (b) how disruptions to and changes in abdominal core function over time affect patient outcomes.

This study has three objectives: (1) to establish the efficacy of standardized post-operative PT to improve function and patient-reported outcomes after ventral hernia repair, (2) the establish the feasibility of this registry-based RCT protocol to test the effect of PT to improve functional and clinical outcomes in individuals undergoing ventral hernia repair, and (3) the determine whether baseline abdominal core function influences the efficacy of post-operative physical therapy after ventral hernia repair. We hypothesize that (1) patients randomized to a standardized PT protocol will achieve larger gains in function and patient-reported function than those randomized to standard-of-care activity limitations, (2) that retention rates will be >80% in both treatment groups, and that compliance rates will be >80% in the PT group, and (3) that those in the PT group with poorer core function at enrollment will achieve greater improvements in function and self-reported function from enrollment to 10 weeks post-operative than those with better core function.

## Methods

This study is a registry-based randomized controlled trial (NCT05142618: Pilot Trial of Abdominal Core Rehabilitation To Improve Outcomes After Ventral Hernia Repair

(ABVENTURE-P)). The protocol has been approved by the Ohio State University Biomedical Institutional Review Board (2021H0336). It will leverage data captured for patients undergoing hernia repair within the Abdominal Core Health Quality Collaborative along with key additional functional measures added for this trial. The study design is consistent with NIH criteria for a Phase I/II trial, in which we are seeking initial safety and efficacy data to inform revisions to research methods and intervention protocols for a future definitive Phase III clinical trial. The schedule of enrollment, interventions, and assessments is shown in Fig 1. The experimental intervention is the post-operative care following ventral hernia repair. Participants will be randomized to receive either (1) standard of care post-operative instructions alone or (2) standard of care post-operative instructions with an 8-week standardized physical therapy protocol [8] focusing on abdominal core function.

Function will be primarily assessed using the Five Times Sit-to-Stand (5xSTS) [9], and patient-reported outcomes using the Patient-Reported Outcomes Measurement Information System Physical Function Computer Adaptive Test (PROMIS-PF-CAT) [10, 11]. The primary endpoint is 10 weeks post-operative at the conclusion of rehabilitation. The outcomes of the trial will be reported using Consolidated Standards of Reporting Trials (CONSORT) recommendations. Sample size estimates for this pilot and feasibility trial were based on preliminary data. Primarily, the aim is to collect a large enough sample to reasonably represent the variability in the population of individuals undergoing ventral hernia repair to power a future definitive multi-center randomized controlled trial.

Ninety-four adult men and women will be enrolled in the study. With 47 patients in each treatment group, we will have 80% power to detect a 20% reduction (i.e., improvement) in 5xSTS scores at 10 weeks in the PT group compared to the control group, assuming coefficient of variation (CV) = 35% and two-sided alpha of 0.025. With that sample size, we will also have 80% power to detect a 40% reduction in PROMIS-PF-CAT scores at 10 weeks in PT group compared to the control group, assuming CV = 100% and two-sided alpha of 0.025.

To evaluate study retention rates and treatment adherence, we will use descriptive statistics. The number of participants in each treatment group who participate in testing at 10 weeks post-operative (both treatment groups). Adherence rates in the PT group will be measured by session attendance, documented by the treating physical therapist. The retention rate and adherence rates will be calculated with 95% confidence intervals using exact binomials for each treatment at each follow-up. With 47 patients in each group, we will be able to estimate the retention rates or compliance rates with 95% confidence intervals of 11.4% if the retention or adherence rate is at least 80% in each treatment group.

Finally, we will use a correlation analysis to best address the hypothesis of whether baseline abdominal core function (as measured by the Quiet Unstable Sitting Test (QUeST) [12]) indicates which patients are most likely to benefit from post-operative rehabilitation versus standard of care. With 47 patients in the PT group, we will have 80% power to detect a Spearman's rank correlation of at least 0.45 between QUeST score and the change of 5xSTS / PROMIS-PF-CAT with a significance level of 0.025.

Ninety-four individuals between the ages of 18 and 70 who are diagnosed with ventral hernia during their initial physician evaluation (e.g., through physical examination, CT scans, ultrasounds) and are scheduled for ventral hernia repair (VHR) within the Division of General and Gastrointestinal Surgery at The Ohio State University will be recruited for this trial (Table 1). The group of patients invited to participate in this study will be ambulatory individuals who are not presenting emergently and are initially seen in an outpatient clinic and deemed suitable candidates for elective VHR. The baseline study visit will occur following the standard outpatient pre-operative visit. Patients will be recruited from a busy General Surgery practice at The Ohio State University. Adjuncts to facilitate recruitment will include email

| | STUDY PERIOD | | | | | | |
|---|---|---|---|---|---|---|---|
| | Enrollment | Allocation** | Post-allocation | | | | Close-out |
| TIMEPOINT | -10 wks* | 0 | 2 wks | 30d | 10 wks | 6mo | 12mo |
| **ENROLLMENT:** | | | | | | | |
| **Eligibility screen** | X | | | | | | |
| **Informed consent** | X | | | | | | |
| **Allocation** | | X | | | | | |
| **INTERVENTIONS:** | | | | | | | |
| *Physical Therapy* | | | | ●━━━━━━━● | | | |
| *Precautions Only* | | | ●━━━━━━━● | | | | |
| **ASSESSMENTS:** | | | | | | | |
| *Five Times Sit to Stand* | X | | | X | X | X | X |
| *PROMIS-Physical Function Computer-Adaptive Test* | X | | | X | X | X | X |
| *Quiet Unstable Sitting Test* | X | | | X | X | X | X |
| *Hernia-Related Quality-Of-Life Survey* | X | | | X | X | X | X |
| *Hernia Recurrence Inventory* | X | | | X | X | X | X |
| *Timed Up and Go* | X | | | X | X | X | X |
| *International Physical Activity Questionnaire – Short Form* | X | | | X | X | X | X |
| *Tampa Scale of Kinesiophobia* | X | | | X | X | X | X |
| *Hernia Width* | | X | | | | | |
| *Compliance with Post-operative Self-Care Guidelines Survey* | | | | X | | | |

**Fig 1. Schedule of enrollment, interventions, and assessments.** *Enrollment occurs when surgery is scheduled. If enrollment is >3 months before surgery, a repeat baseline is performed <10 weeks before surgery. **Allocation occurs the day of surgery after the measurement of the participant's hernia width (i.e. inter-rectus distance).

**Table 1. Subject inclusion and exclusion criteria.**

| Inclusion Criteria | Exclusion Criteria |
|---|---|
| Ages 18–70 | Previously diagnosed movement or balance disorder |
| Diagnosis of ventral hernia | Use of ambulatory assistance device (walker or cane) |
| Scheduled for elective ventral hernia repair | Currently undergoing physical therapy or other skilled exercise intervention supervised by a medical rehabilitation professional at the time of pre-surgical functional measurements |
| Independent functional status | |
| Transverse hernia width of 2cm or greater | |

engagement, social media presence, in-clinic advertising, and electronic health record engagement.

## Recruitment method and screening procedures

Potential participants will be recruited based on findings from their medical evaluation. We aim to enroll 6 participants a month, which will enable us to reach full trial recruitment of 94 participants within 16 months.

Several recruitment strategies will be used to ensure enrollment goals are met. The trial will be advertised on The Ohio State Wexner Medical Center's Center for Abdominal Core Health webpage and posted in clinics with Quick Response (QR) codes so that patients can easily access basic study information and contact details for research personnel. These advertisements will clearly note participant reimbursement for testing and the chance to receive free physical therapy. Individuals presenting to outpatient general surgery clinics for ventral hernia repair (VHR) will be screened by the study team ahead of their appointment. At the time of their scheduled appointment, those who meet all eligibility criteria will be approached by our CRT (e.g., clinical research managers (CRMs), clinical research coordinators (CRCs), and clinical research assistants (CRAs)) regarding participation in the proposed research. Details of the data collection methods and expected time commitment will be clearly described. We will emphasize with each potential participant that this is a clinical trial, in which they will be randomly assigned to one of two groups, one of which receives 16 visits of physical therapy at no cost to them. We will highlight flexibility in scheduling testing and treatment days and times, as appropriate. Members of the CRT will then obtain informed consent from potential trial participants.

The number and proportion of recruited patients from the total number of (a) patients seen in the clinic, (b) patients with ventral hernia screened for eligibility, and (c) patients determined to be eligible for participation will be recorded. The number and proportion of eligible participants who choose to enroll in the study will also be documented.

## Randomization and allocation

On the day of surgery, participants will be randomized (1:1) to one of two treatment arms. Randomization will be stratified by hernia width measured during the procedure using the European Hernia Society Width categories (W1 <4cm, W2 4-10cm, W3 >10cm) [13] to ensure equitable distribution of disease severity between the two treatment arms. The primary study statistician will generate the randomization list using a randomized permuted block scheme. The block sizes will not be known to the PIs. Randomization will take place through the Research Electronic Data Capture (REDCap™) system [14].

We will record the number and proportion of enrolled participants who (a) are randomized, (b) undergo VHR, and (c) receive the assigned treatment (PT vs no PT).

## Blinding

Due to the nature of the two treatment arms, neither participants, treating physical therapists, or treating surgeons can be blinded to group assignment. Study personnel involved in participant testing, the primary statistician (LW), and the blinded MPI (AC) will remain blinded to group assignment. Blinded outcomes assessment following surgery will be performed by the blinded members of the CRT. Participants will be reminded that they should not discuss their treatment with the outcomes assessment team when those assessments are scheduled, when reminder messages are sent, and at the beginning of every assessment. The unblinded MPIs (SD, BP) and unblinded members of the CRT can hold closed meetings when necessary to discuss issues related to specific participants to avoid revealing allocations to the blinded outcomes assessment team. A second statistician is available to provide any necessary unblinded statistical review at the request of the DSMB or the unblinded MPIs.

## Intervention

Those assigned to the intervention group will receive an evidence-based post-operative abdominal core surgery rehabilitation program delivered by physical therapists with experience in treating this patient population. The therapy protocol has been developed and standardized over 5 years through a subset of institutions within the ACHQC and is publicly available via the ACHQC website [8]. A detailed version of the protocol which includes sample exercises, tips for physical therapists to use to elicit proper technique, optimal dosage parameters, and criteria for progression or regression are presented in S1 Appendix.

The overarching goal of the program is to enable pain-free functional mobility and return to prior level of function through targeted exercise and movement re-education. Progression through the phases of the program is criterion-based, guided by time and patient response. The time frames provided are anticipated, not required, and were based on the expert opinions of the clinicians who helped develop the protocol. The program begins with self-directed abdominal wall protection, practicing proper postures and body mechanics with daily activities, walking, and active range of motion with diaphragmatic breathing to prevent complications such as deep vein thrombosis or pneumonia (Weeks 0–2). Supervised physical therapy begins between Weeks 2–4, with goals to learn proper breathing and abdominal and pelvic floor muscle bracing techniques, use proper body mechanics with basic daily activities, and to increase walking tolerance to 30 minutes. Instructions, precautions, and exercises prescribed during the physical therapy intervention will supersede those provided in the standard of care post-operative instructions. In Weeks 4–8, the goals are to improve cardiovascular endurance, tolerate graded exposure to stress at the surgical site to promote tissue adaptation, and gradually return to normal activities of daily living. In Weeks 8+ the physical therapist guides the patient through individualized, progressive resistance training and abdominal and trunk muscle conditioning to achieve optimal muscle function. Patients with poor tolerance to land-based therapy are referred to aquatic physical therapy. In all cases, progression only proceeds when the patient can safely complete the activity on their own without increased pain; the abdomen does not significantly protrude during exercises; and the patient is able to breathe appropriately, i.e. exhale with exertion, avoiding Valsalva [15].

Sixteen physical therapy visits (expected twice per week for 8 weeks) as needed will be paid for by the trial, so no study participants will need to pay out of pocket if they are randomized to the intervention arm. Physical therapists will follow criterion-based guidelines for exercise

progression and discharge. This will allow participants who achieve acceptable outcomes prior to 16 visits to be discharged, as is consistent with ethical care practices. For participants who need additional physical therapy visits beyond what the trial can support, the participant's insurance will be billed. The total number of visits and progress by participants through the standard protocol will be tracked for later examination and will inform the rehabilitation dosage considerations for the future definitive trial.

The control group will receive standard of care post-operative instructions. These include limiting strenuous physical activity and a lifting restriction of no more than 10 pounds for 6 weeks. Binder use is also encouraged for 4–6 weeks after the operation. Both treatment groups will receive standard of care post-operative instructions. These include limiting strenuous physical activity and a lifting restriction of no more than 10 pounds for 6 weeks. Binder use is also encouraged for 4–6 weeks after the operation.

## Relevant concomitant care and interventions that are permitted or prohibited during the trial

During the period between surgery and primary outcomes testing at 10 weeks post-operative, participants in both treatment arms are asked to abstain from additional supervised exercise or physical therapy. This trial permits the treating physical therapists to select other interventions they deem necessary for the care of the participant, which may include but not be limited to manual mobilization of the surgical scar or surrounding tissue and aquatic therapy. Our electronic health record physical therapy documentation template prompts the physical therapist to add this information each visit, and this information will be extracted and summarized at the completion of the trial. In further support of the aims of this feasibility study, we will ask participants to report what kind additional supervised exercise or physical therapy they completed, if any, by self-report survey at the 10 week follow-up. We also anticipate that participants in both groups may elect to undergo new or additional treatments after that timepoint. We will collect these data on additional treatment via participant survey and electronic health record and document rates of cross-over (control group) and continuation of physical therapy (PT group).

## Primary outcome measures

**PROMIS-PF-CAT.** The PROMIS-PF-CAT assesses physical function and includes a validated short form and computer-aided test version [10]. In orthopedic trauma patients who have a wider range of physical function limitations than hernia patients, the CAT version is extremely reliable (Cronbach $\alpha$ = 0.98), takes only 44 seconds to complete, and demonstrates no ceiling or floor effects [10, 16]. The output of the PROMIS-PF-CAT (0–100) is scaled based on the US population, where 50 represents physical function for the average adult, lower scores indicate worse physical function, and higher scores indicate better physical function. Use of PROMIS-PF is novel in abdominal surgery, with only one published study that included patients undergoing various abdominal procedures [11], but its responsiveness to change at one week, three weeks, and five weeks after surgery combined with its known validity in measuring physical function make it the best self-reported measure to use for this RCT.

**Five times sit-to-stand (5xSTS).** The 5xSTS is a common functional test recommended for falls prediction in older individuals [9, 17–19]. It has been used in mobility-impaired individuals such as those with low back pain, hip osteoarthritis, Parkinson's disease, and stroke. The 5xSTS begins and ends with the participant sitting in a standardized chair without arms. The test begins when the participant rises from the chair to a full standing position and sits back down five times, as quickly as possible, without using their arms for support. Timing begins at the participant's first movement and ends when the buttocks contact the chair after

the fifth rise. This test was chosen for the hernia population as it requires bending at the waist, strength, and endurance, all which challenges the abdominal core. In patients with abdominal core dysfunction, we expect that the inability to rise from a chair presents significant limitations on independent mobility in activities of daily living.

**QUeST core stability score.**  We will use the quantitative, continuous abdominal core function assessment (QUeST) at primary and intermediate timepoints. The QUeST is an adapted version of unstable sitting paradigms originally developed to study motor control and muscle activation patterns in individuals with low back pain by several research groups [20–27]. These paradigms have used either a rigid hemisphere or springs mounted to the underside of the chair to make the seat unstable. We adapted this paradigm to the clinic setting by using a BOSU® trainer placed on top of a force-measuring platform (Bertec BP-5046) [12, 28]. The participant sits quietly with eyes closed and arms crossed for 3 trials of 60s each, and counts backwards by 4's to a 60-bpm metronome as a cognitive dual-task to reduce conscious focus on their balance. The QUeST score is then calculated based on the excursion of the center of pressure over each 60s period relative to established normative values [12].

## Secondary outcomes

**Hernia-related quality-of-life survey (HerQLeS).**  The HerQLes is an overall quality of life Likert-style questionnaire that assesses the impact of VHR on abdominal wall function, including activities of daily living, independent mobility, and mental health. The survey has been validated in VHR patients and shown to improve with time after VHR as expected [29]. The HerQLes is included in the ACHQC registry, so every consenting patient who receives a VHR at Ohio State University will have HerQLes data at multiple timepoints during treatment including the timepoints for this study without any effort needed by the research team.

**Hernia recurrence inventory (HRI).**  The HRI has been validated in both inguinal and ventral hernia populations to assess recurrence 1 year or greater after repair [30, 31]. Utilization of the HRI allows a patient reported outcome assessment of recurrence without involving a physical clinical visit with a physician. This secondary outcome of hernia recurrence will be assessed 1 year after the operation.

**Timed Up and Go (TUG).**  The TUG is a common functional test recommended by the Centers for Disease Control and Prevention [32], and often used in mobility-impaired individuals such as those with low back pain [33, 34], total knee arthroplasty [35, 36], and frailty [37–39]. Robust sets of normative values across many different populations exist [40–44]. The TUG begins and ends with the participant sitting in a chair. On command, the patient rises from the chair, walks 3 meters, turns 180˚, walks back, and sits back in the chair. Patients will be asked to perform this as quickly and safely as they can. It requires nothing except a chair, stopwatch, and 4 meters of open hallway, so it is easy to implement in a clinic setting. We have chosen this functional test for the hernia population because it requires bending at the waist, walking, and turning, so it incorporates challenges to mobility both directly and indirectly attributable to the hernia repair itself.

**International physical activity questionnaire—Short form (IPAQ-SF).**  Physical activity level may affect recovery from hernia repair. Therefore, we will collect self-reported physical activity from all participants with the IPAQ-SF which includes four intensity levels: vigorous-intensity activity (e.g., aerobics), moderate-intensity activity (e.g., leisure cycling), walking, and sitting. The IPAQ-SF is reliable with published normative data for individuals in the same age range as the target population for this trial [45]. We will not stratify participants in this pilot and feasibility trial based on physical activity, but we remain cognizant that this stratification may be necessary in a future trial.

**Tampa scale of kinesiophobia (TSK-11).** Kinesiophobia may be a significant driver of poor physical function and quality of life common to those with hernia disease. However, it has never been measured using a standard scale in this population. Therefore, we include the TSK-11 as a measure to help us better understand the role of kinesiophobia as a covariate affecting quality of life and responsiveness to PT intervention, and consider as a candidate for inclusion in future multi-center trials. It has good reliability and responsiveness [46].

Use of Standard Operating Procedures (SOPs) will ensure protocol adherence, and rigorous data collection and management processes. All research personnel will receive training specific to their roles in the clinical trial by the MPIs prior to their involvement in the study using the Manual of Operational Procedures (MOP) and SOPs. Standardized electronic templates for data input, cleaning, and export will be used and regular data audits will be performed. Clinical and survey data are collected in the Abdominal Core Health Quality Collaborative (ACHQC) national registry (i.e. HerQLes, HRI), and we will follow their current processes for data entry and export. Participants will also answer demographic questions and complete additional surveys via REDCap™; the Clinical Research Coordinator will ensure surveys are complete and also use REDCap™ to record functional performance testing. The unblinded MPIs will perform a second data quality check of data entered into REDCap™ within 48 hours. The Ohio State University Wexner Medical Center is the ACHQC Data Coordinator Center and implements standard data assurance review procedures across ACHQC clinical sites. Both ACHQC and REDCap™ maintain rigorous processes for collecting, housing, and exporting coded data. Only approved research team members will have access to these restricted data sets. Physical therapy data will be collected through The Ohio State University Wexner Medical Center's electronic health platform (EHR), Epic. Only physical therapists who have been trained in trial protocol implementation will provide care for trial participants. Standardized templates will be used to record details of the exercises and education provided, allowing discrete data export for data fidelity checks. Smartphrases will be developed and approved for clinician use to ensure ease and consistency in reporting visit details, including participants' response to and progress through the physical therapy protocol. Reliability checklists will be used to assess compliance and consistency with protocol implementation for all participants receiving physical therapy. All participants' charts will be reviewed by an unblinded physical therapist team member at least once during the 10-week intervention period, ensuring data fidelity and allowing retraining and remediation for documentation processes as needed.

Retention will be recorded by the research team at the 30-day, 10-week, 6-month, and 1-year testing sessions, and will include reasons for drop-out (e.g. participant moved out of the area prior to 10-week follow-up). At least eighty percent retention is expected in both treatment groups at the primary endpoint (i.e., 10-week follow-up) [56]. To track adherence, we will collect (a) clinician-documented attendance at physical therapy sessions (PT group) and (b) participant-reported compliance with post-operative self-care guidelines over the 10-week treatment period, including lifting restrictions, brace or binder wear, and restrictions on strenuous vigorous physical activity (both groups).

## Statistical analysis

Statistical analysis will be performed independently by the blinded biostatistician. An intention to treat approach is planned for all analyses, using all participants who were randomized who have available data at the 10-week timepoint. Where appropriate, normality will be assessed, and non-parametric approaches will be used when assumptions are violated. Demographics, clinical characteristics (e.g. age, sex distribution, BMI, baseline pain), and all the secondary outcomes (HerQLes scores, HRI, TUG, IPAQ-LF, and TSK-11) will be summarized for the

physical therapy and control arms, respectively, and compared between the groups using two-sample t-tests for continuous outcomes or Chi-square/Fisher's exact test for discrete outcomes. Linear mixed effect models will be used to study the effect of PT over all the timepoints for those outcomes with longitudinal measures, while including other covariates (biological sex, age, BMI, CDC wound class, IPAQ). The overall type I error will be controlled at 0.05 using the Bonferroni method. We will consider controlling for possible confounders based on the results of our group demographic comparisons. Interaction (treatment group × time) will also be tested in the model. Adjustments for multiple comparisons will be performed.

To evaluate the effect of treatment group allocation on 5xSTS and the PROMIS Physical Function, we will use linear mixed effect models over all the timepoints (10 weeks, 30 days, 6 months, and 1 year). The overall type I error will be controlled at 0.05 using the Bonferroni method. We will consider controlling for possible confounders based on the results of our group demographic comparisons. Interaction (treatment group × time) will also be tested in the model. Adjustments for multiple comparisons will be performed.

To evaluate study retention rates and treatment adherence, we will calculate the number of participants in each treatment group who participate in testing at 10 weeks post-operative (both treatment groups). Adherence rates in the PT group will be measured by session attendance, documented by the treating physical therapist. The retention rate and adherence rates will be calculated with 95% confidence intervals using exact binomials for each treatment at each follow-up. For our last objective, we will use a correlation analysis to best address the question of whether baseline abdominal core function (as measured by QUeST) indicates which patients are most likely to benefit from post-operative rehabilitation versus standard of care.

## Monitoring

The study design is consistent with NIH criteria for a Phase I/II trial in which initial safety and efficacy data will inform revisions to the research methods and intervention protocol for the future definitive Phase III clinical trial. An independent Data and Safety Monitoring Board (DSMB) was convened to assess the progress of the clinical study, the safety data, and critical efficacy endpoints and provide recommendations to the principal investigators, institutional review board. The DSMB is made up of three individuals of relevant expertise to the trial, including experts in ventral hernia repair, clinical trial design and data analysis, and physical therapy trials, with no conflict of interest with the study team members. The Data Safety and Monitoring Plan (DSMP) was approved on prior to the enrollment of the first participant, adjudicated by the Ohio State University's Center for Clinical and Translational Science.

As part of the DSMP for this feasibility randomized controlled trial, we will implement safety stopping rules using strict, a priori criteria, to allow trial termination due to significant harm. We do not anticipate a higher incidence of adverse events (AEs) or serious adverse events (SAEs) in our study participants than is reported in the ventral hernia repair literature due to functional performance testing or participation in physical therapy. We also do not anticipate that participants randomized to receive PT will have higher adverse event rates than those not randomized to PT. Thus, stopping the trial would require detecting a meaningful difference in these AE/SAE rates between the no PT vs PT groups.

We will monitor the rate of adverse events for each of the treatment arms (PT vs no PT) using the stopping criteria outlined below (Table 2). We will minimize the risk that patients will experience AEs in the PT arm by providing standardized physical therapist training on criterion-based exercise progressions and implementing early stopping rules (Table 2). The incidence rate of surgical site infections is 5–9% and the incidence rate of surgical site occurrence

**Table 2. Stopping criteria based on observation of adverse events per treatment arm.**

| Number of participants who have begun the treatment: | Halt if number of unique participants with adverse events is greater than or equal to: |
|---|---|
| 5 | 2 |
| 10 | 3 |
| 15 | 5 |
| 25 | 7 |
| 35 | 9 |
| 45 | 10 |

requiring procedural intervention is 5%. Therefore, we wish to ensure that the adverse events rate defined above is no greater than 15%. The following table gives stopping criteria based on Bayesian Toxicity Monitoring with a posterior probability of stopping for excessive toxicity based on current data $Prob(\theta > 15\% | r,n) \geq 90\%$ [6].

## Discussion

This study protocol is designed to assess the efficacy and feasibility of standardized post-operative PT to improve function and patient-reported outcomes after ventral hernia repair. Standards of practice for many common orthopaedic conditions (e.g. joint replacement) now include physical rehabilitation to maximize beneficial outcomes after surgery. Only recently has the potential value of rehabilitation for individuals following VHR been recognized. The central premise of this trial is that patients undergoing ventral hernia repair will have measurably improved functional performance and quality of life following PT that addresses the mechanism of both the loss and gain of abdominal core function relative to the standard of care (post-operative precautions).

There are some limitations and potential pitfalls to consider with the study design. The most relevant pitfall is inadequate recruitment of participants. To avoid this, we have chosen the inclusion criteria based on a historical volume of patients to ensure we will meet recruitment targets. We have also incentivized the study with physical therapy provided at no cost to participants and gift card compensations at the 10-week, 6-month, and 1-year data collection timepoints. Another pitfall could be loss to follow-up; however, there is a plan in place to minimize this with patient incentives via gift card compensations and an intent-to-treat analysis. Similarly, there may be failure to comply with the physical therapy program. We will use an intent-to-treat analysis to address this pitfall and will report the number of physical therapy sessions attended as part of our aim to assess feasibility. Providing physical therapy at no cost to the participants is also intended to reduce non-compliance.

Upon successful completion of this study, we will establish initial efficacy of pre-operative and post-operative physical therapy to improving patient quality of life, independence, mobility, and function after this procedure. Data from this trial will provide the necessary preliminary data evaluating the effects of the intervention as well as feasibility of scaling this approach for a future, well-powered definitive multicenter clinical trial to reduce the medical and societal burden of ventral hernia on the US population. Data from this trial will establish whether abdominal core function predicts responsiveness to PT and provide robust preliminary data on all primary variables of interest (Table 3) in patients undergoing ventral hernia repair. Using these preliminary data, we will be able to choose the most appropriate variables based both on relevance in the population and feasibility of collection in busy

**Table 3. Primary and secondary variables of interest.**

| | Variables of Interest (VOI) | Description | Measurement properties | Rationale |
|---|---|---|---|---|
| **PRIMARY** | Five Times Sit to Stand (5xSTS) | Time (nearest 0.1 s) taken to stand up and sit down from a standard height chair five times | Reliable (ICC = 0.91–0.96) and sensitive to change (SEM = 0.5–0.9 s) in older sedentary populations;[17, 47–49] MCID = 2.3 s; [50] established age-based norms [51] | Widely used objective physical function test across wide range of conditions, requires use of abdominal core, feasible in clinic setting |
| | PROMIS-Physical Function Computer-Adaptive Test (PROMIS-PF-CAT) | Adaptive questionnaire from 120-item bank to assess self-reported physical function | Reliable (Cronbach α = 0.98), takes 44 s to complete, no observable floor or ceiling effects, scores 0–100 [10] | Validated physical function measure across wide range of ability, responsive to change after abdominal surgery [11] |
| | Quiet Unstable Sitting Test (QUeST) | Assessment of abdominal core function using postural sway while seated on unstable surface, eyes closed, cognitive dual-task (counting down) | Reliable (Intra-rater ICC = 0.88) [28] with initial validity in hernia patients established | Objective, more precise measure of integrated function of all major muscle groups in abdominal core, no observable floor effects |
| **SECONDARY** | Hernia-Related Quality-of-Life Survey (HerQLes) | 12-item survey of quality of life with Likert-style scale appropriate for patients with hernia disease | Reliable (Rasch person reliability statistic = 0.86) and valid in adults undergoing ventral hernia repair [29] | Commonly used to assess change in quality of life following ventral hernia repair. *Standard measure in ACHQC national registry*. |
| | Hernia Recurrence Inventory (HRI) | Two-item self-reported questionnaire asking about appearance of a bulge, pain/symptoms at surgical site | 85% sensitive, 81% specific to detect recurrence. Patients reporting bulge 18x more likely to have a recurrence (95% CI 3.7–90.0) [30] | Commonly used to assess hernia recurrence, doesn't require imaging. *Standard measure in ACHQC national registry*. |
| | Timed Up and Go (TUG) | Time (nearest 0.1 s) to stand up from a standard height chair, walk 3 m, turn 180˚, return and sit down in chair | Reliable (ICC = 0.97) and sensitive to change (SEM = 0.21 s) in adults with lumbar degenerative disc disease; [52] MCID = 3.4 s; [53] established age-based norms [18] | Widely used objective physical function test across wide range of conditions, requires use of abdominal core, feasible in clinic setting |
| | International Physical Activity Questionnaire–Short Form (IPAQ-SF) | Self-reported physical activity questionnaire | Repeatability coefficient of ρ = 0.81 (95% CI 0.79–0.82) and 0.84–1.00 percent agreement in categorical activity classification [45] | IPAQ-SF is appropriate for individuals ages 15–69, participating in a wide range of physical activity. |
| | Tampa Scale of Kinesiophobia (TSK-11) | 11-item self-reported questionnaire asking about fear of movement and physical activity participation due to pain. | 11-items (scores between 11–44 points), with good reliability and responsiveness (ICC = 0.81; SEM 2.54, MCID = 4 points) [46] | Commonly used to assess fear in populations with movement impairment [54, 55] |

clinical practice to accurately estimate needed sample sizes for the multi-center randomized control trial. Further, data from this trial will establish a detailed blueprint for collaboration between surgery and physical therapy to successfully execute the planned multi-center clinical trial.

## Supporting information

**S1 Appendix. Detailed physical therapy protocol.** The protocol includes sample exercises, tips for physical therapists to use to elicit proper technique, optimal dosage parameters, and criteria for progression or regression.
(PPTX)

**S1 Checklist. SPIRIT 2013 checklist: Recommended items to address in a clinical trial protocol and related documents\*.**
(DOC)

**S1 Data.**
(DOCX)

## Acknowledgments

We would like to express our sincere gratitude to the physical therapists from the Sports Medicine and Ambulatory physical therapy clinics at the Ohio State University Wexner Medical Center (OSUWMC) who volunteered to serve as interventionists on this trial. We value their expertise and willingness to spend extra time completing required training to work with our participants. We also acknowledge the enthusiasm and commitment of the participating surgeons at OSUWMC, whose collaboration with our clinical research team members (Kiana Shannon and Mahsa Adib) will ensure that interested and eligible patients have the opportunity to participate in this study.

## Author Contributions

**Conceptualization:** Stephanie Di Stasi, Ajit M. W. Chaudhari, Lai Wei, Benjamin K. Poulose.

**Data curation:** Stephanie Di Stasi, Ajit M. W. Chaudhari, Savannah Renshaw, Lai Wei, Laura Ward.

**Formal analysis:** Stephanie Di Stasi, Ajit M. W. Chaudhari, Lai Wei.

**Funding acquisition:** Stephanie Di Stasi, Ajit M. W. Chaudhari, Lai Wei, Benjamin K. Poulose.

**Investigation:** Stephanie Di Stasi, Ajit M. W. Chaudhari, Savannah Renshaw, Lai Wei, Laura Ward, Elanna K. Arhos, Benjamin K. Poulose.

**Methodology:** Stephanie Di Stasi, Ajit M. W. Chaudhari, Savannah Renshaw, Lai Wei, Laura Ward, Elanna K. Arhos, Benjamin K. Poulose.

**Project administration:** Stephanie Di Stasi, Ajit M. W. Chaudhari, Savannah Renshaw, Elanna K. Arhos, Benjamin K. Poulose.

**Resources:** Stephanie Di Stasi, Ajit M. W. Chaudhari, Benjamin K. Poulose.

**Supervision:** Stephanie Di Stasi, Ajit M. W. Chaudhari, Benjamin K. Poulose.

**Writing – original draft:** Stephanie Di Stasi, Ajit M. W. Chaudhari, Elanna K. Arhos, Benjamin K. Poulose.

**Writing – review & editing:** Stephanie Di Stasi, Ajit M. W. Chaudhari, Savannah Renshaw, Lai Wei, Laura Ward, Benjamin K. Poulose.

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
