## [Decision Letter · Decision Letter 0]

9 May 2023

PONE-D-23-01179Pilot Trial of Physical Therapy Versus Standard of Care Following Ventral Hernia Repair: Protocol for a Randomized Controlled TrialPLOS ONE

Dear Dr. Chaudhari,

Thank you for submitting your manuscript to PLOS ONE. After careful consideration, we feel that it has merit but does not fully meet PLOS ONE’s publication criteria as it currently stands. Therefore, we invite you to submit a revised version of the manuscript that addresses the points raised during the review process.

We look forward to receiving your revised manuscript.

Kind regards,

Renato S. Melo, PhD

Academic Editor

PLOS ONE

“ This study is supported by funding from the National Institute of Diabetes and Digestive and Kidney Diseases (NIDDK) (R01DK131207)[56]. This project also receives in-kind administrative support by The Ohio State University Center for Clinical and Translational Science, which receives financial support from the National Center for Advancing Translational Sciences (NCATS) (UL1TR002733) and by the Abdominal Core Health Quality Collaborative (ACHQC). Neither NIDDK nor NCATS assisted in the design of the trial and will not contribute to data collection and management, analysis and interpretation, dissemination of trial findings, and does not have authority over these activities. The content is solely the responsibility of the authors and does not necessarily represent the official views of the National Institutes of Health or the ACHQC.”

“This study is funded by the National Institute of Diabetes, Digestive and Kidney Diseases (https://www.niddk.nih.gov) grant R01DK131207 (SLDS, AMWC, BKP PI's). This project also receives in-kind administrative support by The Ohio State University Center for Clinical and Translational Science, which receives financial support from the National Center for Advancing Translational Sciences (NCATS) (UL1TR002733) and by the Abdominal Core Health Quality Collaborative (ACHQC). Neither NIDDK nor NCATS assisted in the design of the trial and will not contribute to data collection and management, analysis and interpretation, dissemination of trial findings, and does not have authority over these activities. The content is solely the responsibility of the authors and does not necessarily represent the official views of the National Institutes of Health. The funders had and will not have a role in study design, data collection and analysis, decision to publish, or preparation of the manuscript.”

“I have read the journal's policy and the authors of this manuscript have the following competing interests: Benjamin K. Poulose receives salary support as Vice President of the Board of the Abdominal Core Health Quality Collaborative and is an equity holder in EndoEvolve, LLC.”

5. Please upload a copy of Figure 5, to which you refer in your text on page 14. If the figure is no longer to be included as part of the submission please remove all reference to it within the text.

Additional Editor Comments:

Authors should follow reviewer requests as soon as possible.

Reviewers' comments:

Reviewer's Responses to Questions

**Comments to the Author**

1. Does the manuscript provide a valid rationale for the proposed study, with clearly identified and justified research questions?

Reviewer #1: Yes

Reviewer #2: Yes

2. Is the protocol technically sound and planned in a manner that will lead to a meaningful outcome and allow testing the stated hypotheses?

Reviewer #1: Yes

Reviewer #2: Yes

3. Is the methodology feasible and described in sufficient detail to allow the work to be replicable?

Reviewer #1: Yes

Reviewer #2: No

4. Have the authors described where all data underlying the findings will be made available when the study is complete?

Reviewer #1: Yes

Reviewer #2: Yes

5. Is the manuscript presented in an intelligible fashion and written in standard English?

Reviewer #1: Yes

Reviewer #2: Yes

6. Review Comments to the Author

You may also provide optional suggestions and comments to authors that they might find helpful in planning their study.

Reviewer #1: This is a well-written protocol of a feasibility phase I/II trial of physical therapy versus standard of care for hernia. Randomization procedures are clearly specified and are well thought out. Goals of the study are clear and achievable. Statistical analysis plan is adequate. I have some comments:

1. What about the use of APPS to report daily symptoms and analyze this as a continuous time series?

2. Your goal is feasibility, determination of sample size for a larger study, preliminary efficacy on some metrics. You should have a separate section on what specific measures and criteria will be used to make a decision about moving to a larger study, and to specifically inform the design of such a study.

3. Limitations and how to handle stuff that doesn't work out should be added to the Discussion.

Reviewer #2: Thank you for the opportunity to review this manuscript. Some points need to be clarified better. Are they:

1. Regarding the introduction, the current state of evidence on the subject of the study needs to be better defined and outlined.

2. About the inclusion criteria: How will the diagnosis of ventral hernia be made?

3. On the description of interventions: Interventions can be better described and defined. They are understandable. However, as it is a protocol, these can be better described.

7. PLOS authors have the option to publish the peer review history of their article (what does this mean?). If published, this will include your full peer review and any attached files.

Reviewer #1: No

Reviewer #2: No

---

## [Author Response · Author response to Decision Letter 0]

14 Jun 2023

Response to reviewers is attached as a file.

---

## [Decision Letter · Decision Letter 1]

10 Jul 2023

ABVENTURE-P pilot trial of physical therapy versus standard of care following ventral hernia repair: protocol for a randomized controlled trial

PONE-D-23-01179R1

Dear Dr. Chaudhari,

We’re pleased to inform you that your manuscript has been judged scientifically suitable for publication and will be formally accepted for publication once it meets all outstanding technical requirements.

Kind regards,

Renato S. Melo, PhD

Academic Editor

PLOS ONE

Additional Editor Comments (optional):

Dear authors, we are pleased to inform you that the manuscript entitled: ABVENTURE-P pilot trial of physical therapy versus standard of care following ventral hernia repair: protocol for a randomized controlled trial, has been accepted for publication in PLos One.

Congratulations once again and thank you for choosing to publish your valuable studies in our journal.

Reviewers' comments:

Reviewer's Responses to Questions

**Comments to the Author**

1. Does the manuscript provide a valid rationale for the proposed study, with clearly identified and justified research questions?

Reviewer #1: Yes

Reviewer #2: Yes

2. Is the protocol technically sound and planned in a manner that will lead to a meaningful outcome and allow testing the stated hypotheses?

Reviewer #1: Yes

Reviewer #2: Yes

3. Is the methodology feasible and described in sufficient detail to allow the work to be replicable?

Reviewer #1: Yes

Reviewer #2: Yes

4. Have the authors described where all data underlying the findings will be made available when the study is complete?

Reviewer #1: Yes

Reviewer #2: Yes

5. Is the manuscript presented in an intelligible fashion and written in standard English?

Reviewer #1: Yes

Reviewer #2: Yes

6. Review Comments to the Author

You may also provide optional suggestions and comments to authors that they might find helpful in planning their study.

Reviewer #1: All comments have been addressed adequately xxxxxxxxxxxxxxxxxxxxxxxxxxxxxxxxxxxxxxxxxxxxxxxxxxxxxxxxxxxxxxxxxxxxxxxxxxxxxxxxxxxxxxxxxxxxxxxxxxxxxxxxxxxxxxxxxxxxxxxxxxxxxxxxxxxxxxxxxxxxxxxxxxxxxxxxxxxxxxxxxxxxxxxxxxxxxxxxxxxxxxxxxxxxxxxxxxxxxxxxxx

Reviewer #2: I congratulate the authors for the review work. As the revision carried out the manuscript can be accepted.

7. PLOS authors have the option to publish the peer review history of their article (what does this mean?). If published, this will include your full peer review and any attached files.

Reviewer #1: No

Reviewer #2: No

---

## [Editor Report · Acceptance letter]

18 Jul 2023

PONE-D-23-01179R1 

ABVENTURE-P pilot trial of physical therapy versus standard of care following ventral hernia repair: protocol for a randomized controlled trial 

Dear Dr. Chaudhari:

I'm pleased to inform you that your manuscript has been deemed suitable for publication in PLOS ONE. Congratulations! Your manuscript is now with our production department. 

Kind regards, 

on behalf of

Dr. Renato S. Melo 

Academic Editor

PLOS ONE